# Extending the Design Life of the Palm Jumeirah Revetment Considering Climate Change Effects

Khaled Elkersh [1] , Serter Atabay [1],* , Abdullah Gokhan Yilmaz [2] , Yomna Morad [1] and Nour Nouar [1]

[1]   Department of Civil Engineering, American University of Sharjah,
     Sharjah P.O. Box 27272, United Arab Emirates; b00073468@alumni.aus.edu (K.E.);
     g00072919@alumni.aus.edu (Y.M.); g00074390@alumni.aus.edu (N.N.)
[2]   Department of Engineering, La Trobe University, Melbourne, VIC 3086, Australia; g.yilmaz@latrobe.edu.au
*    Correspondence: satabay@aus.edu

**Abstract:** This paper presents potential upgrades to the Palm Jumeirah Island's outer revetment to extend its design life for 50 years, considering the sea level rise (SLR) associated with climate change. The paper proposes several upgrade options to ensure that the hydraulic stability and wave overtopping discharges of the Palm Jumeirah revetment comply with the recommended design criteria based on industry guidelines. The performance of the existing revetment, in terms of the hydraulic stability and wave overtopping discharge criteria, is assessed using design wave heights (1- and 100-year events) extracted from an extreme wave analysis study on the Dubai coast. The results show that, based on the new design conditions, the existing structure should be upgraded to meet the armor stability criteria and recommended overtopping discharge values. Three different upgrade solutions are designed and analyzed to satisfy the required hydraulic stability and overtopping conditions. The suggested upgrade options are an extra armor layer, a flat berm, and a submerged breakwater offshore. The proposed upgrade solutions are preliminary designs that would require verification in terms of their geotechnical stability and physical model testing to evaluate their performance.

**Keywords:** revetment; climate change; sea level rise




## 1. Introduction

Coastal defense structures are constructed to dissipate and reflect incident wave energy to protect the coast against wave effects. These structures include seawalls, groynes, and breakwaters and their hydraulic stability is essential for avoiding damage or a loss of functionality [1]. With the rise in sea levels caused by climate change, these structures need to be upgraded to comply with the original performance criteria, which includes stability and overtopping [2–4]. Coastal structures are usually upgraded by reinforcing or modifying the existing structure's geometry, or even by adding extra structural elements.

Coastal areas are vulnerable to the threats of climate change [4–8]. The main impact of climate change is sea level rise (SLR), which is caused by the melting water of glaciers and the thermal expansion of seawater as it warms. This rise in sea level puts existing coastal structures at risk of accelerated damage and failure. Hence, Toimil et al. (2020) argued that traditional coastal engineering practices need to be adjusted to face these climate change threats [9]. A study conducted by Esteban et al. (2013) examined the impacts of four different SLR scenarios to quantify the increase in the size of the existing breakwater to accommodate the constantly changing climate conditions [10]. The authors argued that the increase in sea level will influence the cost of building rubble mound structures. Similarly, the cost of breakwaters designed in the future will increase compared to those designed in the 20th century; however, their designed service life needs to be extended. Furthermore, the authors suggested that it is likely that the design philosophy of coastal structures must change to consider future uncertainties in the wave climate [10].

Artificial manmade islands are examples of coastal structures that are prone to failure due to strong wave action. Hellebrand et al. (2004) presented a case study on the existing design of a revetment that protects an artificial manmade island, Palm Jumeirah, from the threat of natural disturbances [11]. This revetment design considered factors including the wave's height and energy to protect the island's beaches and small marinas from erosion and reduce the hydrodynamic loads on the coastal infrastructure. Therefore, in this study, the revetment of Palm Jumeirah Island is assessed to determine if its existing revetment complies with the set performance criteria under climate change effects and defines several upgrade solutions for the revetment to maintain a long service life.

There are several approaches to upgrading a rock-armored revetment. Burcharth et al. (2014) and Eldrup et al. (2019) conducted design exercises to examine the possible structure upgrades for an existing rock-armored revetment through the modification or addition of structural elements [4,12]. The predictions for sea level rise, as well as the increase in long-term wave conditions, were taken into consideration in the study. The upgrade scenarios for the revetment were cost-optimized to select the most feasible and economical upgrades. Similarly, Koftis et al. (2015) presented a methodological approach to upgrading emerged and submerged rubble-mound breakwaters [13]. Their paper explored various upgrade designs with their corresponding cost estimates, concentrating on climate change effects. The upgrades were studied for their hydraulic and structural responses and the selection criteria were based on the construction costs. These studies provide multiple valuable upgrade options, which will serve as the base references for the upgrade options in this study on the Palm Jumeirah Island revetment.

This paper aims to investigate the performance of the existing rubble-mound structure located at Palm Jumeirah, Dubai, United Arab Emirates (UAE), by analyzing its hydraulic stability and wave overtopping considering the rise in sea level. The parameters investigated are the stability of the slope and toe, overtopping, and the filter criteria. Hence, along with assessing the existing Palm Jumeirah rubble-mound revetment with the updated wave and SLR conditions, this paper studies and recommends multiple upgrade solutions for the existing structure that would extend its design life by 50 years. These solutions include changes in the geometry of the existing structure, in addition to an offshore breakwater structure. This paper uses the data provided in the original design of the revetment by Hellebrand et al. (2004) to assess the existing Palm Jumeirah revetment [11].

## 2. Materials and Methods

### 2.1. Study Area

The manmade Palm Jumeirah Island is an artificial island shaped like a date palm tree that extends into the Arabian Gulf, a semi-enclosed ocean that is, on average, 800 km long and 200 km wide [14]. Figure 1 shows a satellite image of Palm Jumeirah Island at the coordinates of 25°6′44.64″ N, 55°8′20.4″ E, which was extracted from Google Earth 7.3 (2022). The island has a 1.5 km long trunk and 17 fronds with a crescent-shaped revetment that frames the island to protect these fronds from waves. The 11 km long revetment was constructed to protect the manmade island's beaches and coastline from strong waves originating from the Arabian Gulf. There are two gaps at each side of the crescent-shaped revetment to ensure water exchange with the open water and around the fronds of the palm tree. The gaps also allows a passageway for recreational vessels in and out of the island. A rubble-mound armor protects the seaside of the revetment, while the inside is used as a beach consisting of unprotected sand [11]. The original cross-section of the revetment is displayed in Figure 2, with all the displayed measurements in millimeters. The levels described in this figure are with respect to the Dubai Municipality Datum (DMD). This profile has an armor layer sloping at 1:2 using 3–6 t rocks, in addition to a toe and crest made up of 1–3 t rocks. Furthermore, a wide crest is utilized to collect the overtopping discharge and transport it back into the sea [11].

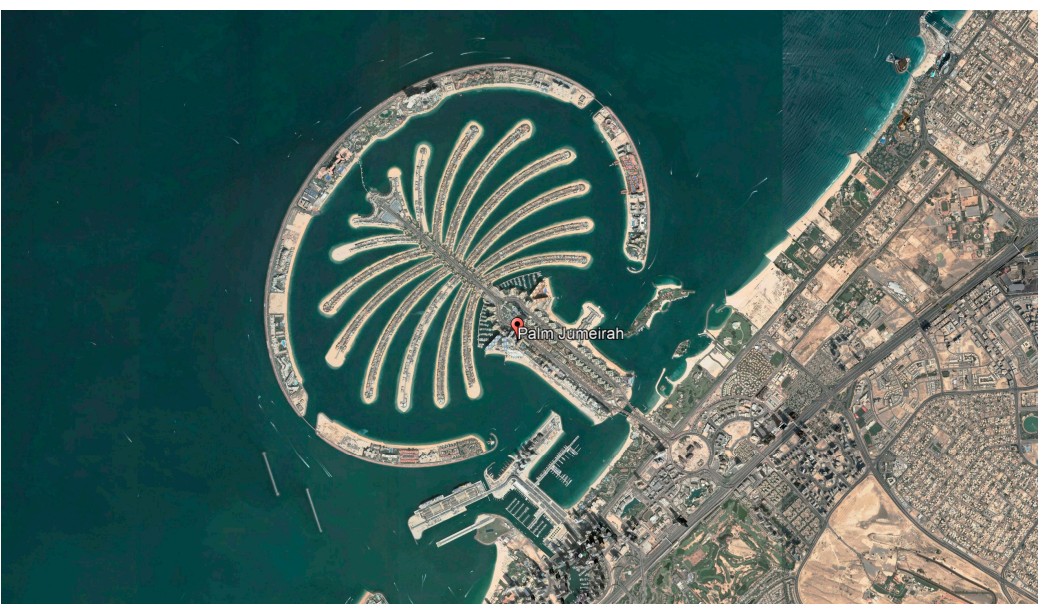

**Figure 1.** Palm Jumeirah Island from Google Earth 7.3, 2022.

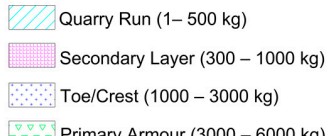

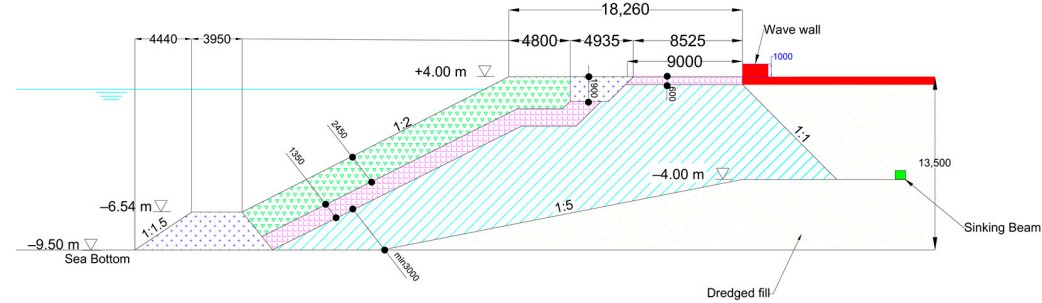

**Figure 2.** Original revetment cross-section (in mm if not specified).

## 2.2. Setting of Definitions

### 2.2.1. Service Lifetime

Coastal structures that are relatively permanent, such as rock-armored revetments, are typically designed for a service lifetime or design life of 50 years. This study aims to extend the design life of the revetment of Palm Jumeirah Island by 50 years, considering the climate change effects on sea level rise. Hence, the analysis and design of the structure consider the estimated sea level rise scenarios described in Section 2.3.

### 2.2.2. Performance Criteria

The performance criteria are set in order to ensure the safety and effectiveness of the rubble-mound revetment in withstanding the hydraulic forces generated by waves. This study has identified the following performance criteria for the Palm Jumeirah revetment:

- Armor stability: the damage level is set as the initiation of armor damage, with $S_d = 2$ in Van Der Meer's formula for the 100-year return period event.
- Toe stability: the damage level is set for the toe stability as the start of damage, with $N_{od} = 0.5$.

- Filter criteria: to ensure the long-term stability of the revetment, the filter criteria limits are defined as found in the Rock Manual [15].
- Overtopping criteria: overtopping is the average discharge rate of water along the breakwater per linear meter. In this study, the overtopping criteria at the wave wall of the revetment for the 1-year and 100-year return events are set as 0.03 L/s/m and 1.00 L/s/m, respectively, based on the original design criteria.

### 2.2.3. Revetment Crest Level

The crest level of the revetment is designed not to exceed +4.0 m (DMD) or consider "no increase to crest level". This is the existing revetment's crest level and it was proposed to avoid blocking the sea view.

### 2.2.4. Design Wave Heights

Forty-year hindcast wave data covering the period of 1979–2018 were originally obtained to analyze and create models for the wave action along the Dubai coast, United Arab Emirates [16]. An extreme wave analysis is performed on this wave data to model the significant wave heights at different return periods [17]. The authors use the Peak Over Threshold method to filter the wave data by selecting a threshold of 2.5 m, which yields around 113 storms per year. Only the north-west direction is filtered, as it is considered to be the most significant wave direction due to Shamal storms, which are yearly recurrent storms that move in the south-east direction over the course of a few days [11]. The study suggests that the generalized Pareto distribution (GPD) fits the wave data the best compared to other extreme wave distributions. Table 1 presents the results of the study, showing the wave heights ($H_{m0}$) and corresponding peak periods ($T_p$) for the 1-year and 100-year return periods [17].

**Table 1.** Wave heights and peak periods for the 1- and 100-year return periods [17].

| Return Period (Years) | $H_{m0}$ (m) | $T_p$ (s) |
|:---:|:---:|:---:|
| 1 | 3.00 | 8.7 |
| 100 | 4.37 | 10.3 |

### 2.3. Climate Change Scenarios

The effects of different climate change scenarios are defined based on the Intergovernmental Panel on Climate Change (IPCC). Considering the project design life of 50 years, different SLR scenarios are investigated around the year 2070. SLR values of 0.3, 0.33, 0.34, and 0.42 m are considered, which correspond to the mean SLR values in 2070 for the 2.6, 4.5, 6.0, and 8.5 RCP scenarios, respectively [18].

### 2.4. Upgrading

The existing revetment has several potential upgrade options, including but not limited to:

- Adding an extra armor layer
- Adding an extra armor layer with a milder slope
- Adding a flat berm
- Adding a submerged breakwater offshore

## 3. Results

### 3.1. Assessment of the Existing Structure

The existing Palm Jumeirah revetment is studied and analyzed using the new wave conditions and set design criteria. The analysis of the revetment begins with using the SwanOne standalone, which is a one-dimensional Graphical User Interface for Swan that transforms the offshore wave conditions to nearshore conditions at the revetment [19]. In this study, the distance between the offshore wave data point and the revetment is

approximately 2540 m. The bed level at the wave data point is −11.1 m and −9.5 m at the revetment, which suggests a low seabed slope of 1/1600. The offshore bathymetry is assumed to be represented by parallel bottom contours to the average coastline. Using SwanOne, four design water level models are created: the design high water levels (DHWL) and design low water levels (DLWL) for the 1- and 100-year return periods. The boundary conditions are different for the different models created. The water depth is the summation of the surge, SLRs for the different climate change scenarios, and mean higher high water levels (MHHW). For instance, for the 100-year return period's DHWL, the water depth includes an MHHW of +1.7 m (DMD), SLRs for the different climate change scenarios, and a surge of 1 m, whereas for the DHWL for the 1-year return period, a storm surge of 0.4 m is used instead. For the DLWL, the water depth is taken as the Lowest Astronomical Tide (LAT = 0). The adopted tidal datums in this study are based on historical observations of tidal levels.

The four different climate change scenarios show insignificant variability in the created SwanOne models. Therefore, a single RCP scenario is used in this study. The model created for the RCP8.5 scenario, with an SLR of 0.42 m, is extracted, as shown in Table 2. The significant wave height ($H_{m0}$) of the DHWL for the 100-year return period is found to be 4.23 m, as seen in Table 2. However, in the original design of the Palm Jumeirah revetment, the 100-year significant wave height considered in the design is 4.0 m [11]. Therefore, it is evident that the existing revetment should be reassessed for its armor and toe stability, overtopping, and filter.

**Table 2.** Swanone models (RCP8.5).

| Model | DHWL 1-Year | DLWL 1-Year | DHWL 100 Years | DLWL 100 Years |
|---|---|---|---|---|
| Water level | +2.52 | 0 | +3.12 | 0 |
| Spectral significant wave height—$H_{m0}$ (m) | 2.86 | 2.86 | 4.23 | 3.66 |
| Peak period—RTP (S) | 8.96 | 8.96 | 10.0 | 10.0 |
| Mean absolute wave period—TM01 (S) | 6.9 | 6.2 | 7.0 | 6.3 |
| Period based on first negative moment of energy spectrum—TMM10 (S) | 7.6 | 7.0 | 8.0 | 7.5 |
| Direction spreading of waves—DSPR (Deg) | 30 | 30 | 30 | 27 |
| Water depth—DEPTH (m) | 12.02 | 9.50 | 12.63 | 9.58 |
| Mean wave height of the highest 1/3rd of the waves—$H_{1/3}$ (m) | 2.91 | 2.96 | 4.39 | 3.78 |
| Wave height exceeded by 2% of the waves—$H_{2\%}$ (m) | 4.08 | 3.82 | 5.47 | 4.60 |

### 3.1.1. Armor Stability

The armor layer of the structure is the outer layer of the revetment, which is directly exposed to the incident sea waves. Van Der Meer's formula, which is shown in Equation (1), is used in this study for plunging the waves for the armor layer stability from the Rock Manual [15]. The formula is used to determine the minimum mean diameter ($D_{n50}$) and minimum mean weight ($M_{50}$) of the rocks needed for the structure to be stable against a significant wave height of 4.23 m, which corresponds to the 100-year DHWL, as shown in Table 2. The calculated $D_{n50}$ and $M_{50}$ are approximately 1.45 m and 8.1 t, respectively. However, the rock grading used for the existing armor layer of the Palm Jumeirah revetment is 3–6 t, which is smaller than the minimum calculated rock diameter and weight that would satisfy the stability of the armor layer. Hence, the revetment's armor layer does not comply with the armor stability criteria at the 100-year event and upgrade solutions should be implemented.

$$\frac{H_s}{\Delta D_{n50}} = c_{pl} * P^{0.18} * \left(\frac{S_d}{\sqrt{N}}\right)^{0.2} * \xi m^{-0.5} \tag{1}$$

where $D_{n50}$ is the nominal mean diameter of the armor units (m), $S_d$ is the damage number taken as 2 (start of armor damage), $N$ is the number of incident waves at the toe, $H_s$ is the significant wave height (m), $\xi m$ is the surf similarity parameter using the mean wave

period, $\alpha$ is the slope angle, $\Delta$ is the relative buoyant density, $P$ is the notional permeability of the structure, taken as 0.4, and $c_{pl}$ is an empirical coefficient taken as 6.2.

### 3.1.2. Toe Stability

The toe of a revetment is the part of the structure that is in contact with the seabed and it is responsible for providing a stable foundation for the structure. In this study, the toe stability is tested for the 1-year DLWL and 100-year DLWL using Equation (2) [20]. Table 3 shows the calculated rock diameter and weight that would satisfy the stability of the revetment's toe. In this case, the values calculated for the 100-year DLWL condition govern, as they are higher than the calculated values for the 1-year DLWL condition. Currently, the rock class designation used in the existing revetment's toe is 1–3 t with a $D_{n50}$ of 1.22 m, which is acceptable in terms of hydraulic stability, as it satisfies the set stability criteria of the toe without any adjustments.

$$\frac{H_s}{\Delta D_{n50}} = \left(2 + 6.2\left(\frac{h_t}{h}\right)^{2.7}\right) N_{od}^{0.15} \tag{2}$$

where $D_{n50}$ is the nominal mean diameter (m), $h$ is the water depth in front of the toe (m), $h_t$ is the water depth at the structure toe (m), and $N_{od}$ is a constant taken as 0.5, which corresponds to "almost no damage".

**Table 3.** Toe stability.

| Scenarios | $H_s$ (m) | $h_t$ (m) | $h$ (m) | $D_{n50}$ (m) | $M_{50}$ (kg) |
|---|---|---|---|---|---|
| 1-year | 3.0 | 6.5 | 9.5 | 0.50 | 333 |
| 100-year | 3.8 | 6.6 | 9.6 | 0.58 | 537 |

### 3.1.3. Overtopping

One of the design requirements for the revetment at Palm Jumeirah is to limit the overtopping volumes that pass or might cause flooding hazards in the areas it protects. Hence, it is vital to analyze the revetment for overtopping when considering the sea level rise. Two different approaches are utilized in determining the overtopping discharge rate. The first approach involves using the empirical overtopping formula, Equation (3), obtained from the EurOtop manual [21].

$$\frac{q}{\sqrt{g * H_{m0}^3}} = 0.09 \times exp\left(-\left(1.5 \frac{R_c}{H_{m0}} \frac{1}{\gamma_f}\right)^{1.3}\right) \tag{3}$$

where $q$ is the mean overtopping rate (L/s/m), $g$ is the acceleration due to gravity (m/s$^2$), $H_{m0}$ is the spectral significant wave height (m), $R_c$ is the crest freeboard (m), and $\gamma_f$ is the roughness factor. According to the EurOtop Manual, the empirical formula is used only for simple straight slopes. For the existing revetment, the armor layer consists of two rock layers with a permeable core. Therefore, a roughness factor ($\gamma_f$) of 0.40 is used in the overtopping discharge equation. As more energy can be dissipated in a wide crest, a reduction factor ($Cr$) with a maximum of $Cr = 1$ is calculated using Equation (4) and applied to the overtopping discharge ($q$).

$$Cr = 3.06 \times exp\left(-1.5\frac{G_c}{H_{m0}}\right) \tag{4}$$

where $G_c$ is the crest width (m) and $H_{m0}$ is the spectral significant wave height (m). The overtopping discharge rates are calculated as 0.002 and 1.63 L/s/m for the 1-year and 100-year DHWLs, respectively. Based on the obtained overtopping rates, the 1-year DHWL

value is within the set overtopping criteria, but the 100-year DHWL value exceeds the overtopping discharge limit.

The second approach involves using the artificial neural network (ANN) simulations tool (overtopping.ing.unibo.it) adopted by the EurOtop manual [21–23]. The ANN tool includes more complex geometries and overcomes most of the limitations of the traditional empirical formulae, which should provide more accurate estimates. Furthermore, the ANN tool uses algorithms and data collection to study different scenarios to calculate the overtopping discharge. In this study, the output of the ANN simulations provides average overtopping discharge rates of 0.16 and 2.05 L/s/m for the 1-year and 100-year DHWLs, respectively. Based on the obtained overtopping rates, the 1-year and 100-year DHWL values exceed the set overtopping discharge limits. The empirical formula and ANN simulations yield different overtopping values. The difference between the outcomes of these methods is due to the different inputs and considered parameters used in each method. However, both methods suggest that the existing revetment cross-section needs to be adjusted accordingly to satisfy the overtopping criteria.

### 3.1.4. Filter Design

The filter design of the revetment plays a significant role in ensuring the stability and longevity of the structure. In this section, the existing revetment of Palm Jumeirah is checked against the filter criteria set in the Rock Manual to ensure migration, interface stability, internal stability or uniformity, and permeability [15]. Table 4 presents the characteristics of the rock gradings used in the existing revetment. The filter calculations presented in Table 5 show that the existing revetment satisfies the filter criteria limits of the Rock Manual.

**Table 4.** Rock grading characteristics.

|  | Armor (3–6 t) | Filter (0.3–1 t) | Core (1–500 kg) |
|---|---|---|---|
| *NLL* | 3000 | 300 | 1 |
| *NUL* | 6000 | 1000 | 1000 |
| $M_{50}$ | 5800 | 671 | 64 |
| $D_{10}$ | 1.28 | 0.61 | 0.12 |
| $D_{15}$ | 1.32 | 0.64 | 0.15 |
| $D_{60}$ | 1.47 | 0.78 | 0.40 |
| $D_{85}$ | 1.54 | 0.84 | 0.61 |
| $D_{n50}$ | 1.22 | 0.63 | 0.29 |

**Table 5.** Filter criteria analysis.

| Filter Criteria | Armor (3–6 t) | Filter (0.3–1 t) | Core (1–500 kg) | Limits |
|---|---|---|---|---|
| Migration, $M_{50f}/M_{50b}$ | 6.92 | - | - | <15–20 |
| Interface stability, $D_{15f}/D_{85b}$ | 1.57 | 1.05 | - | <5 |
| Internal stability, $D_{60}/D_{10}$ | 1.15 | 1.28 | 3.33 | <10 |
| Permeability, $D_{15f}/D_{15b}$ | - | 4.27 | - | >1 |

### 3.2. Proposed Solutions

Based on the analysis performed on the existing revetment, the structure needs to be upgraded in order to comply with the set design criteria. Based on Van Der Meer's formula, which was used to check the stability of the slope, the armor layer does not comply with the stability criteria. Additionally, the overtopping criteria values are not satisfied by the existing revetment. However, the toe of the structure is stable and the filter criteria are satisfied. Therefore, three upgrade solutions are proposed to extend the design life of the revetment of Palm Jumeirah by satisfying the stability, filter, and overtopping requirements.

### 3.2.1. Solution 1: Extra Armor Layer with a Milder Slope

The first upgrade solution involves adding armor rock units on top of the existing structure at a milder slope, as shown in Figure 3. This solution is designed to ensure that the armor layer complies with the armor stability criteria by referring to Van Der Meer's formula [15]. In this solution, the notional permeability coefficient in the formula is increased from $P = 0.4$ to $P = 0.6$, as suggested by Burchart et al. (2014), to consider the effect of an extra armor layer on the armor stability [4]. The slope at the existing revetment armor layer would not satisfy the armor stability criteria. Therefore, when using the same rock size used in the existing armor layer (3–6 t), the slope should be at least 1:2.5 to maintain a stable armor layer. In order to comply with the overtopping criteria, the empirical overtopping formula in Equation (3) is used to calculate the overtopping rates, by increasing the crest width to reduce these overtopping rates. A crest width increase of 2.44 m is sufficient to reduce the overtopping to 0.001 and 0.69 L/s/m to meet the set 1-year and 100-year DHWL overtopping criteria, respectively. The proposed increase in the crest width is equivalent to the width of two rock layers at the top of the extra layer, with the same rock grading as the existing armor layer. The existing toe of the structure shows hydraulic stability; hence, it is extended with the same rock grading. Similarly, the filter design for this upgraded solution complies with the criteria, as is the case in the existing revetment.

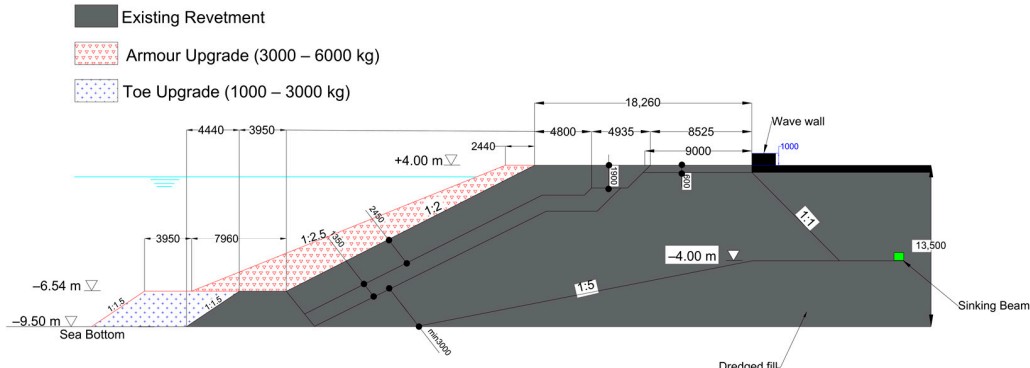

**Figure 3.** Extra armor layer with a milder slope solution.

### 3.2.2. Solution 2: Flat Berm

The second upgrade solution involves adding a flat berm on top of the existing armor layer to comply with the armor stability and overtopping criteria, as shown in Figure 4. However, adding the berm with the existing slope does not satisfy the armor stability criteria, as is the case in the existing revetment. Therefore, Van Der Meer's formula is utilized to calculate the minimum required slope for the rocks to maintain hydraulic stability [15]. Like solution 1, the calculated minimum required slope to maintain the stability of the revetment is 1:2.5 by adopting a notional permeability coefficient of 0.6. However, the berm geometry, which includes the width and slope of the berm, as well as the distance from the top of the berm to the still water level, is further investigated based on the overtopping discharge rates obtained using the ANN overtopping tool. The ANN overtopping tool is used as it is better at simulating the complex geometry of the upgraded revetment than the empirical formula. This approach suggests that, in order to satisfy the overtopping limits, the berm should be 4.88 m wide and slope at 1:3.5. This increase in the crest width is equivalent to four rock layers with the same rock grading as the existing armor layer. The simulation provides average overtopping discharge rates of 0.03 and 0.15 L/s/m for the 1-year and 100-year DHWLs, respectively, which comply with the set overtopping criteria for both return periods. The toe of the existing structure is extended as it satisfies the toe stability. The filter design for this upgrade option complies with the criteria, as is the case in the existing structure. Finally, the stability of the upper slope is checked using the Van Gent formula for upper slope stability [24]. The formula suggests

that the upper slope of the revetment with the berm satisfies the stability criteria without any modifications.

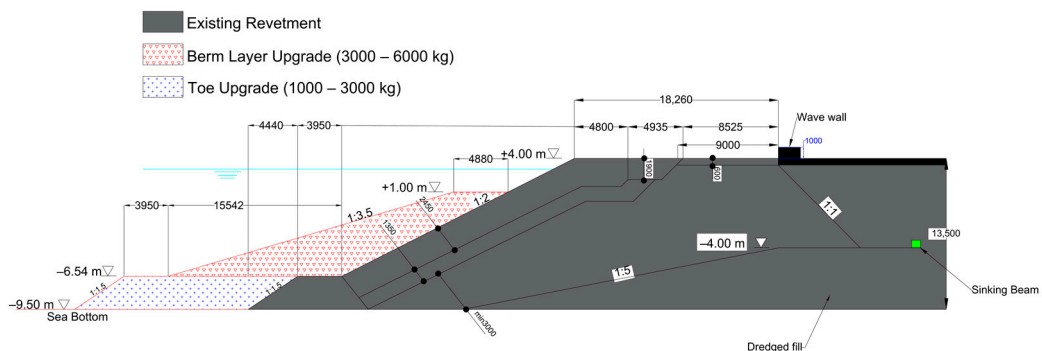

**Figure 4.** Flat berm solution.

### 3.2.3. Solution 3: Submerged Breakwater on the Foreshore

The third upgrade solution consists of designing a submerged breakwater to be constructed offshore to dissipate the wave energy and decrease the incoming wave height. The maximum allowable $H_s$ at the existing structure, which complies with the set armor stability, is calculated to be 3.24 m using Van Der Meer's formula. The wave transmission ($C_t$) is calculated to be 0.77, which is the ratio of the transmitted significant wave height (3.24 m) to the incident significant wave height (4.23 m). The geometry of the submerged breakwater is calculated using the equations in the Rock Manual as follows [15]. The freeboard needed ($R_c$) is calculated to be −3.35 m using Equation (5) and the crest level is calculated to be −0.2 m by adding the calculated $R_c$ to the DHWL (3.04 m).

$$C_t = 0.46 - 0.3\frac{Rc}{Hs} \tag{5}$$

The height of the submerged breakwater would then be equal to the distance from the sea bottom to the crest level, which is equal to 9.3 m. The stability of the front slope of the armor layer is determined from the Rock Manual as a function of the relative crest height based on the ratio of $R_c/D_{n50}$. The value of $D_{n50}$ is calculated to be 1.20 m, which is a rock grading of 3–6 t. Equation (6) for the wave transmission from the EurOtop manual [21] is used to calculate the width of the submerged breakwater to be 15.2 m.

$$C_t = -0.4\frac{Rc}{H_s} + 0.64\left(\frac{B}{H_s}\right)^{-0.31}(1 - exp(-0.5\xi p)) \tag{6}$$

With the transmitted wave conditions to the existing revetment, an overtopping check is performed for the 1-year and 100-year events using the EurOtop empirical formula in Equation (3), and the overtopping criteria are met. For the filter and core design calculations, a standard grading of 0.3–1 t is chosen for the filter layer and 1–500 kg is chosen for the core layer. By using these gradings, all the filter criteria limits are within an acceptable range, as shown previously in Table 4. Like the armor layer, the filter layer rocks are placed in two layers with layer thicknesses of 1.26 m. The final cross-section of the proposed submerged breakwater solution is shown in Figure 5.

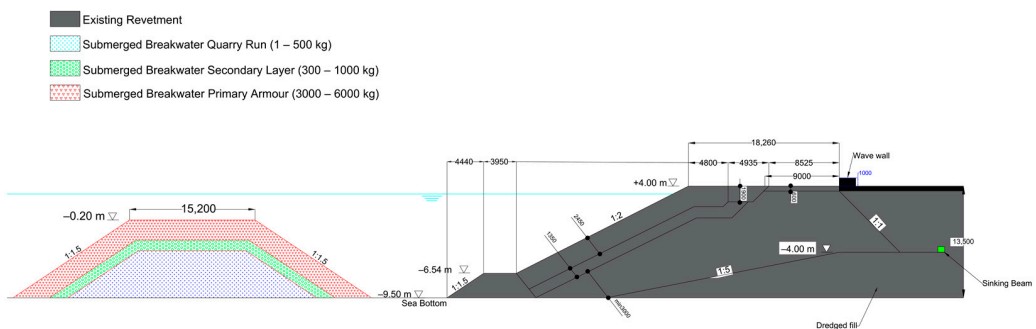

**Figure 5.** Submerged breakwater solution.

## 4. Discussion

In this study, the revetment at Palm Jumeirah Island was analyzed for its hydraulic stability and overtopping. The analysis of the existing revetment showed that, with the considered effects of climate change (RCP 8.5) on the SLR, the existing revetment does not comply with the armor stability and overtopping criteria. However, the revetment complies with the toe stability and filter criteria. Hence, this paper proposed three different upgrade solutions to extend the design life of the revetment by 50 years. The three solutions were designed to ensure the hydraulic stability and satisfy the overtopping criteria. However, further verification is required in terms of their geotechnical stability and physical model testing is required to evaluate their performances. The proposed solutions are an extra armor layer with a milder slope (solution 1), a flat berm (solution 2), or a submerged breakwater (solution 3).

Alternatively, there are other options that could be suitable for rubble-mound revetments. For instance, adding an extra armor layer with the same slope as the existing revetment would serve as a potential upgrade solution. This solution might satisfy the overtopping limits, but not the armor stability criteria. Therefore, in order to obtain a stable armor layer, the size of the armor units used for the extra layer should be larger than those currently used in the existing revetment, which are not available locally. Hence, the extra armor layer should be placed at a milder slope, as suggested in solution 1. Similarly, increasing the crest level by adding an extra armor layer could serve as an upgrade option and would help with reducing the overtopping. However, in this case, the crest level should not exceed the existing revetment's crest level, in order to not block the sea view. Furthermore, as investigated by Gao et al. (2021), a Bragg submerged breakwater would perform well in coastal defense, as would the submerged breakwater option [25].

Unlike solution 3, solutions 1 and 2 show similarities in their required rock gradings and the practicalities of their construction. Therefore, the solutions proposed in this study need to be further investigated in terms of their volumes and estimated costs, in order to select the most economical solution. Additionally, there are other selection factors that should be considered when selecting the optimum solution, such as constructability, environmental and social aspects, and the feasibility of further upgrading the structure in the future.

## 5. Conclusions

This study assessed the performance of the existing revetment located at Palm Jumeirah Island, based on climate change, by assessing its stability, overtopping, and filter criteria. The results showed that, due to the climate change effects on sea level rise, the existing revetment does not meet the stability criteria at the armor level, in addition to having overtopping discharge rates that exceed the recommended overtopping rates. Hence, this paper proposed three different feasible solutions for upgrading the existing revetment. To enumerate, the proposed solutions are: adding an extra armor layer with a milder slope, adding a high berm on top of the existing armor layer, and lastly, adding a submerged breakwater on the foreshore. These upgrades would allow for the structure to increase its stability against the adverse impacts of climate change, in addition to reducing its over-

topping rates. The optimum upgrade solution is the one that satisfies both the functional and economic requirements. Ultimately, it is crucial to invest in upgrading coastal defense structures to increase their resilience against the anticipated effects of climate change and ensure their continued functionality in protecting the shoreline.

**Author Contributions:** Conceptualization, S.A.; methodology, K.E.; validation, K.E., Y.M. and N.N.; formal analysis, K.E.; investigation, K.E., Y.M. and N.N.; resources, S.A.; writing—original draft preparation, K.E.; writing—review and editing, K.E.; visualization, K.E. and A.G.Y.; supervision, S.A.; project administration, S.A. All authors have read and agreed to the published version of the manuscript.

**Funding:** This research received no external funding.

**Institutional Review Board Statement:** Not applicable.

**Informed Consent Statement:** Not applicable.

**Data Availability Statement:** No new data were created or analyzed in this study. Data sharing is not applicable to this article.

**Acknowledgments:** The authors would like to acknowledge Filipe Vieira for his guidance and support throughout the study.

**Conflicts of Interest:** The authors declare no conflict of interest.

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
