# Peer review of "Extending the Design Life of the Palm Jumeirah Revetment Considering Climate Change Effects"

_hydrology, doi:10.3390/hydrology10050111_

Round 1

Reviewer 1 Report

Major revision is suggested. Please see the attachment.

Reviewer 2 Report

This article addresses a relevant issue, which is coastal defense structure redesign due to climate changes. The manuscript is well written and the results are clear. However, I believe some methodological aspects must be enhanced and better explained, as well as text structure in some points.

I suggest the authors to focus in the following topics:

1) Figure 2: please provide the reference level adopted. Horizontal lengths are in milimeters? It is not explained.

2) Lines 105 – 106, section 2.2.1 : AUTHORS: "Hence, the analysis and design of the structure consider the estimated sea level rise values around the year 2070". COMMENT:  the sea level rise values adopted were not presented. This information appears only in section 2.3, lines 137 to 141. Please verify text structure.

3) Lines 108 – 113: The Van Der Meer’s formula, Nod and filter criteria are cited here, in section 2.2.2. However, the text introduces these parameters only after line 174 (Section 2.5.1). Please, verify text structure.

4) Lines 123 - 134: authors shoul better characterize the wave climate in the region. SUGGESTION: please present wave energy distribution by direction, or wave height and peak period distribution by direction (e.g. wave rose figure).

5) Line 139: 0.34 is repeated in the text.

6) Lines 150 – 159: Why SwamOne? It is a 1D model. As stated in the Introductory chapter of the SwanOne User’s Manual: "THE 1D-MODE ASSUMES THAT THE OFFSHORE BATHYMETRY CAN BE REPRESENTED BY PARALLEL BOTTOM CONTOURS SUCH THAT THE BOTTOM PROFILE CAN BE SPECIFIED ALONG ONE TRANSECT NORMAL TO THE (AVERAGE) COASTLINE”

Question: does this assumption applies to the studied region? If it is not, wave refraction and diffraction may not be well represented. This question may be answered by presenting and analysing a figure with the bathymetric contours of the studied coast. Instead, the authors stated only that the bottom slope is mild.

7) - Line 158 - 164: Authors did not explain how the adopted water levels were obtained. For instance: MHHW = +1.71 m; storm surge = 0.4 m. Please provide the source of these information. Field data? Tidal level prediction?

8) - Line 247: "in order" instead of "I order".

9) - Line 305: AUTHORS: “The geometry of the submerged breakwater is calculated using equations in the rock manual” SUGGESTION: please explain how it was performed, instead of only citing the rock manual.

10) Text structure: I understood that Section 2.5. – “Assessment of the Existing Structure” reports and analysis performed by the authors. In this context, I believe this section should be included in the “Results” chapter, since these analyses provided the information required for structure redesign.

This paper is not ready to be published at this moment. However, if the authors perform the required revisions, I believe this article may be acceptable for publication in this special issue of Hydrology.

Reviewer 3 Report

I'm sorry for the delayed review.

The topic is very interesting. The manuscript is interesting to read as example for climate change adaptation. I have only a few minor comments:

Table No. 3 is wrongly numbered.

In table 3 the last column is given in kg. But the second line seems to give the weight in tonns.

In table 2 the significant wave height and peak period is the same in the first two columns DHWL and DLWL despite that have different water levels.

In the first sentence of the introduction I would rewrite. The defense structures may also act to reflect wave energy and thus protect the coast.

Round 2

Reviewer 1 Report

The present version can be accepted.

Reviewer 2 Report

The authors performed all the requested corrections and explained the topics pointed out in my first review. In my opinion, this paper is now ready to be published.